# AI-Automated Risk Operative Stratification for Severe Aortic Stenosis: A Proof-of-Concept Study

**DOI:** 10.3390/jcm14238304

**Published:** 2025-11-22

**Authors:** Dorian Garin, Diego Arroyo, Ioannis Skalidis, Philippe Di Cicco, Charlie Ferry, Wesley Bennar, Serban Puricel, Pascal Meier, Mario Togni, Stéphane Cook

**Affiliations:** 1Department of Cardiology, University and Hospital Fribourg, 1708 Fribourg, Switzerland; dorian.garin@icloud.com (D.G.);; 2Department of Cardiology, Geneva University Hospitals, 1206 Geneve, Switzerland

**Keywords:** large language models, EuroSCORE II, Heart Team, risk stratification, transcatheter aortic valve replacement, artificial intelligence

## Abstract

**Background**: Accurate operative risk stratification is essential for treatment selection in severe aortic stenosis. We developed an automated workflow using large language models (LLMs) to replicate Heart Team risk assessment. **Methods**: We retrospectively analyzed 231 consecutive patients with severe aortic stenosis evaluated by multidisciplinary Heart Teams (January 2022–December 2024). An automated system using GPT-4o was developed, comprising the following: (1) structured data extraction from clinical dossiers; (2) EuroSCORE II calculation via two methods (algorithmic vs. LLM-based); (3) clinical vignette generation; and (4) risk stratification comparing EuroSCORE-based thresholds versus guideline-integrated LLM approaches with/without EuroSCORE values. The primary endpoint was the risk stratification accuracy of each method compared to Heart Team decisions. **Results:** Mean age was 79.5 ± 7.7 years, with 58.4% female. The automated workflow processed patients in 32.6 ± 6.4 s. The LLM-calculated EuroSCORE II showed a lower mean difference from Heart Team values (−1.42%, 95% CI −2.32 to −0.53) versus algorithmic calculation (−1.88%, 95% CI −2.38 to −1.38). For risk stratification, the guideline-integrated LLM without EuroSCORE achieved the highest accuracy (90.0%) and AUC (0.93), outperforming both the EuroSCORE-based (accuracy 50.2% for high-risk, AUC 0.63) and guideline-integrated LLM with EuroSCORE approaches (accuracy 82.4%, AUC 0.76). However, systematic hallucinations occurred for cardiovascular risk factors when data were missing. **Conclusions:** LLMs accurately calculated EuroSCORE II and achieved 90% concordance with multidisciplinary Heart Team decisions. However, hallucinations, reproducibility concerns, and the absence of clinical outcome validation preclude direct clinical application.

## 1. Introduction

Accurate operative risk stratification by multidisciplinary Heart Teams is essential for treatment selection in severe aortic stenosis, with SAVR indicated for low-risk and TAVR for high-risk patients [1,2,3,4]. Current assessment relies on EuroSCORE II to predict 30-day cardiac surgical mortality [5]. However, EuroSCORE II demonstrates poor calibration in contemporary populations, systematically underestimates mortality in high-risk patients, and shows reduced accuracy for TAVR candidates [2,6,7]. Manual risk calculation is time-consuming, error-prone, and challenging to reproduce, particularly when processing large patient cohorts [8,9].

Large language models show promise for cardiovascular risk assessment [10,11,12,13,14]. Open-source local models are increasingly considered for clinical documentation tasks given data protection requirements, though performance varies across architectures [15]. While recent studies have explored LLM applications for cardiovascular risk assessment and Heart Team decision support [16], no prior study has automated the complete pipeline from unstructured clinical documents through structured data extraction, risk score calculation, and guideline-integrated stratification.

This study developed a comprehensive automated workflow utilizing LLM technology to process patient files, calculate EuroSCORE II, and perform guideline-based risk stratification for patients with severe aortic stenosis. We assessed whether LLMs could replicate Heart Team operative risk stratification decisions, and investigated the relationship between model architecture and medical task accuracy.

## 2. Materials and Methods

We conducted a retrospective cohort study of 231 consecutive patients with severe aortic stenosis evaluated by the multidisciplinary Heart Team at the University and Hospital of Fribourg between 1 January 2022 and 4 December 2024. The Heart Team comprised interventional cardiologists, cardiac surgeons, cardiac anesthesiologists, and imaging specialists who met weekly using standardized protocols to discuss all severe aortic stenosis cases. Decisions were made by consensus after multidisciplinary discussion considering operative risk scores, frailty assessment, echocardiographic findings, coronary anatomy, patient preferences, and procedural feasibility. In cases where initial opinions diverged, discussion continued until consensus was reached, and the final consensus decision was recorded as the reference standard for this study.

Inclusion criteria comprised severe aortic stenosis requiring intervention and completion of formal risk assessment during Heart Team evaluation. Data were collected in the Cardio-FR registry (NCT04185285), a single-center cohort initiated in January 2015. The registry was conducted in accordance with the Declaration of Helsinki, received approval from the local ethics committee (003-REP-CER-FR), and written informed consent was obtained from all participants. Institutional Review Board approval was confirmed for this study.

### 2.1. Development of the Automated Clinical Workflow

We developed a four-step automated system to replicate traditional Heart Team evaluation processes (Figure 1). The complete workflow was designed to process patient data from extraction through risk stratification in a single automated pipeline.

**Step 1: Data extraction.** Anonymized Heart Team patient files were processed using optical character recognition (OCR). We evaluated 12 open-source models capable of structured data extraction, utilizing pydantic (Python 3.9.0) to extract predefined variables such as demographics, comorbidities, echocardiographic measurements, coronary anatomy, and laboratory results. All extracted data were manually checked for accuracy. The complete, reusable open-source code for our pipeline is available at https://github.com/garind-888/automated-heart-team-risk-stratification (accessed on 19 September 2025) and is referenced hereafter as Appendix A.

**Step 2: EuroSCORE II calculation.** We implemented two complementary calculation methods to assess accuracy. The first approach used algorithmic calculation applying the official EuroSCORE II formula with standardized parameters reflecting our Heart Team protocols. Patient variables were processed by this algorithm to calculate the EuroSCORE II value. The second approach employed LLM-based calculation, where local models received patient files with instructions to calculate EuroSCORE II using the complete logistic regression formula in a single prompt. Each patient underwent 10 independent calculation iterations to assess reproducibility. The prompt used and algorithmic setup are provided in Appendix A.

**Step 3: Clinical vignette generation.** To provide the final model with the clinical histories of the patients in natural language, another script transformed structured data into standardized clinical narratives mimicking Heart Team case presentations. The template incorporated patient demographics, cardiac history, presenting symptoms, comorbidities, imaging findings, and laboratory results to create consistent narratives suitable for risk assessment. Appendix A contains the template used.

**Step 4: Final risk stratification.** We evaluated two distinct approaches against Heart Team decisions as the reference standard. The EuroSCORE-based approach applied established guideline thresholds where low risk was defined as age less than 75 years and EuroSCORE II less than 4%, while high risk was defined as EuroSCORE II greater than 8%. The Guidelines-integrated LLM approaches processed clinical vignettes using GPT-4o (version 2024–05–13, OpenAI, San Francisco, CA, USA). These approaches used combined prompting strategies, including our previously published tree-of-thought reasoning, to simulate multidisciplinary discussion. The system employed guided reflection and self-consistency through 40 iterations per patient with majority vote determination [17]. We tested implementations both with and without EuroSCORE II values to assess numerical anchoring effects. The model received the aortic stenosis section from the 2021 ESC/EACTS Guidelines for valvular heart disease management. All used prompts are available in Appendix A.

### 2.2. Technical Infrastructure and Implementation

Models were deployed on a local MacBook Pro M4 with 48 GB RAM and 16 cores running macOS Sequoia 15.5 with Ollama version 0.9 for open-source models. GPT-4o version 2024-05-13 was accessed via Microsoft Azure OpenAI Service in Switzerland North, ensuring data residency compliance. Model parameters were standardized with the temperature set to 1.0, top_p to 1.0, frequency_penalty to 0.0, and max_tokens to 4000 to balance creativity with consistency. All system interactions were logged for audit purposes with role-based access control.

### 2.3. Statistical Analysis

The primary endpoint was risk stratification accuracy compared to Heart Team decisions. Secondary endpoints included EuroSCORE calculation agreement, sensitivity, specificity, and area under the receiver operating characteristic curve (AUC). Continuous variables were expressed as mean with standard deviation or median with interquartile range based on distribution. Categorical variables were reported as frequencies and percentages. Friedman’s rank sum test assessed differences among stratification approaches, with post hoc pairwise comparisons using Wilcoxon’s signed-rank test with Bonferroni’s correction. Agreement between calculation methods was evaluated using Bland–Altman analysis and intraclass correlation coefficients. ROC curves were constructed with bootstrap resampling using 1000 iterations for 95% confidence intervals. For open-source model comparison, we calculated mean absolute error (MAE) versus Heart Team EuroSCORE II and assessed correlation between accuracy, defined as 100 minus MAE, and model characteristics using Spearman’s correlation. Statistical significance was set at *p* less than 0.05. Analyses were performed using Python 3.13.0 with SciPy 1.10.1, NumPy 1.24.3, Pandas 2.0.1, and Scikit-learn 1.5.2.

## 3. Results

### 3.1. Study Population Characteristics

The cohort of 231 consecutive patients reflected typical severe aortic stenosis demographics with a mean age of 79.5 ± 7.7 years, and female predominance at 58.4% (Table 1). Mean EuroSCORE II was 4.35 ± 3.15% and STS score was 2.8 ± 1.62%. The Heart Team recommended transcatheter intervention for 168 patients (72.7%) of the cohort, surgical replacement for 50 patients (21.6%), and medical management for 13 patients (5.6%).

### 3.2. Performance of the Automated Workflow

The complete automated workflow achieved a mean processing time of 32.6 ± 6.4 s per patient (Table 2). Manual validation revealed 94.7% overall accuracy across 38 predefined variables, with no significant differences between the extraction models (Friedman’s test, *p* = 0.99). However, we observed systematic and statistically significant over-reporting of cardiovascular risk factors due to OCR interpretation errors. When OCR returned ambiguous checkbox values displayed as “x Yes x No”, all 12 models uniformly hallucinated findings rather than flagging missing data or returning “unknown” values

For hypertension, automated extraction identified 208 patients (90.0%) as hypertensive compared to 170 patients (73.6%) in manual validation, representing a significant overestimation of 38 cases (16.4%, 95% CI 10.1–22.7, *p* < 0.001). For dyslipidemia, automated extraction identified 185 patients (80.1%) versus 110 patients (47.6%) in manual validation, overestimating by 75 cases (32.5%, 95% CI 25.3–39.7, *p* < 0.001).

### 3.3. EuroSCORE II Calculation

Among the tested models, deepseek-r1:14b achieved the lowest mean absolute error at 5.38, followed by qwen3:4b at 5.41 and medllama2:7b at 5.44. Higher MAEs were observed for qwen2.5:14b at 16.49, meditron at 13.07, and gemma3:1b at 15.78. No significant correlation existed between model accuracy and either size in gigabytes or parameter count in billions, as shown in Figure 2. Overall, no intra-patient variability was observed across 10 iterations of score calculation (ANOVA F = 0.42, *p* = 0.92). Compared to Heart Team values, LLM-calculated values were lower by −1.42% (95% CI −2.32 to −0.53, *p* = 0.002). Algorithmic calculation yielded a mean difference of −1.88% (95% CI −2.38 to −1.38, *p* < 0.001). Absolute error was lower for the LLM versus algorithmic calculation (mean difference −1.54, 95% CI −2.39 to −0.70, *p* < 0.001).

### 3.4. Risk Stratification Performance

Risk stratification demonstrated high reproducibility with no significant intra-observer variance across 40 queries for the guideline-integrated LLM approaches (with EuroSCORE: *p* = 0.998; without EuroSCORE: *p* = 0.979). For low versus non-low risk classification, accuracy was 90.05% (95% CI 86.07–94.02) for the guideline-integrated LLM without EuroSCORE, 85.97% (95% CI 81.36–90.59) for the EuroSCORE-based approach, and 82.35% (95% CI 77.29–87.42) for the guideline-integrated LLM with EuroSCORE. For high versus non-high risk classification, accuracy was 90.05% (95% CI 86.07–94.02) for the guideline-integrated LLM without EuroSCORE, 82.35% (95% CI 77.29–87.42) for the guideline-integrated LLM with EuroSCORE, and 50.23% (95% CI 43.58–56.87) for the EuroSCORE-based approach. Pairwise comparisons for low versus non-low risk showed that the LLM without EuroSCORE was superior to the EuroSCORE-based approach (difference 4.07%, 95% CI 0.21–7.93, *p* = 0.039) and to the LLM with EuroSCORE (difference 7.69%, 95% CI 2.82–12.56, *p* = 0.002). For high versus non-high risk, the LLM without EuroSCORE exceeded the EuroSCORE-based approach by 39.82% (95% CI 31.68–47.96, *p* < 0.001) and the LLM with EuroSCORE by 7.69% (95% CI 2.82–12.56, *p* = 0.002). The guideline-integrated LLM without EuroSCORE achieved the highest AUC (0.93), with a sensitivity of 0.98 and specificity of 0.64 (Figure 3). The guideline-integrated LLM with EuroSCORE showed an AUC of 0.76 with a sensitivity of 0.88 and specificity of 0.64. The EuroSCORE-based approach demonstrated an AUC of 0.72 for non-high-risk and 0.63 for high-risk classification (Table 3).

## 4. Discussion

This proof-of-concept study demonstrates two distinct capabilities of LLM-based automation in severe aortic stenosis evaluation. First, LLMs accurately calculated EuroSCORE II with lower mean absolute error compared to algorithmic approaches, representing a purely computational task with objective verification. Second, LLMs achieved 90% concordance with multidisciplinary Heart Team operative risk stratification decisions, representing complex clinical reasoning that integrates multiple factors beyond numerical scores. However, systematic hallucinations from data extraction errors represent critical safety limitations requiring resolution before clinical application.

The 30 s processing time compares favorably to traditional Heart Team preparation, which involves substantial chart review time with risk of human error [18]. This efficiency gain could address workflow inefficiencies identified as primary drivers of clinician burnout [18], though our findings represent a preliminary demonstration rather than an implementation-ready solution. Previous cardiac surgery decision support systems have shown promise in maintaining clinical parameters and reducing medication errors [19], though our system’s integration of multiple complex tasks remains experimental and requires extensive validation before clinical deployment [20].

The finding that the LLM-calculated EuroSCORE II more closely approximated Heart Team values than algorithmic calculation represents an unexpected advantage in this computational task. This aligns with evidence that machine learning approaches can outperform traditional cardiovascular risk scores, with meta-analyses showing ML algorithms achieving AUCs between 0.80 and 0.90 for cardiovascular outcomes [21,22]. The systematic underestimation of risk by both algorithmic and LLM-calculated EuroSCORE II compared to Heart Team values confirms recent validation studies showing poor calibration, particularly for high-risk patients [6,7]. The finding that LLM calculations more closely approximated clinical values suggests that language models may better interpret ambiguous clinical data or variables subject to interpretation, such as “poor mobility,” though this advantage was modest and requires further investigation.

Model accuracy for EuroSCORE II calculation did not correlate with size or parameter count, with smaller models like qwen3:4b performing comparably to larger ones. This suggests domain-specific optimization and structured clinical data may matter more than raw scale for computational tasks, indicating efficient, smaller models could offer similar accuracy with lower resource utilization, though further validation is needed.

Beyond numerical score calculation, operative risk stratification represents a different task requiring integration of clinical judgment, guideline interpretation, and multifactorial decision-making. Excluding EuroSCORE II counterintuitively improved risk stratification accuracy. Possible explanations include anchoring bias from numerical scores constraining holistic assessment [23], limitations in processing mixed numerical and textual inputs, or guideline-based reasoning better capturing multifactorial clinical decision-making [24]. This finding suggests that LLMs may excel at synthesizing qualitative clinical information and guideline recommendations rather than relying on single numerical anchors, though the mechanisms underlying this performance require further investigation.

The systematic hallucination of cardiovascular risk factors represents a fundamental safety concern. Despite explicit prompting to return “NA” or “missing” for absent data, all models uniformly hallucinated positive findings for hypertension and dyslipidemia when OCR returned ambiguous checkbox values. This reflects the inherent tendency of language models to generate plausible content when faced with uncertainty, particularly dangerous in medical contexts where accuracy is paramount [25,26,27]. Recent evidence shows hallucination rates of 1.47% and omission rates of 3.45%, even in controlled settings [28], suggesting that our observed problem may reflect broader challenges in medical AI deployment. Medical hallucinations frequently use domain-specific terms and appear coherent, making them difficult to recognize without expert scrutiny [27]. Human oversight may shift mental effort from content creation to verification, potentially introducing new error risks [29]. For clinical translation, we propose a safe-failure protocol requiring OCR confidence scores above 0.90 for processing to proceed, automatic flagging of contradictory or ambiguous values as “missing data,” structured output schemas forcing selection between defined categories, including “uncertain” options, and calibrated confidence scores for all extracted variables. However, our current implementation lacks automated uncertainty detection capable of identifying ambiguous OCR output or distinguishing hallucination from genuine extraction. Future development must prioritize uncertainty quantification and automatic error detection before clinical application.

Future validation must address multiple dimensions before clinical deployment, with distinct requirements for score calculation versus clinical stratification. For EuroSCORE II calculation, external validation should assess computational accuracy across diverse patient populations and documentation formats. For operative risk stratification, prospective studies should assess whether LLM-based decisions predict actual procedural mortality, morbidity, and long-term survival rather than merely concordance with expert opinion. Multi-center external validation across different Heart Team cultures and documentation styles is essential to establish generalizability, particularly given institutional variations in electronic health record templates, language, and clinical workflows that may require site-specific adaptation of OCR processing and data extraction prompts. Integration of multimodal data, including DICOM imaging and direct echocardiographic parameter extraction, would enhance the system’s comprehensiveness and address current limitations from relying solely on structured clinical vignettes. Minimum dataset requirements should include patient demographics, complete echocardiographic measurements, laboratory values, documented comorbidities, and prior cardiac interventions to ensure reliable pipeline operation. An important research direction involves exploring systems that automatically update their context when ESC/EACTS guidelines are revised or new evidence emerges, though such adaptive systems would require regular human oversight and revalidation to ensure safe incorporation of updated criteria without introducing errors from outdated or misinterpreted recommendations.

Beyond technical validation, implementation requires careful attention to clinical integration, ethical frameworks, and economic considerations. Any deployment must function exclusively as decision support rather than an autonomous decision-maker, with the final responsibility remaining with treating physicians and the Heart Team. Human-in-the-loop safeguards must include mandatory physician review of all LLM-extracted data, risk calculations, and stratification recommendations before clinical decisions, with clear documentation of agreement or disagreement with automated suggestions. The legal and ethical framework for clinical responsibility when using AI-assisted tools remains an evolving area requiring regulatory guidance, institutional policy development, and clear communication with patients about the role of AI in their care [29]. Clinician acceptability studies evaluating trust, workflow integration, and human–AI collaboration dynamics will be essential to understand real-world adoption barriers. Health economic evaluation comparing costs and outcomes of AI-assisted versus traditional workflows must consider evidence that high accuracy does not guarantee cost-effectiveness, with improving AI sensitivity potentially increasing medical costs, while increasing specificity reduces unnecessary referrals but may weaken detection capability [30]. Establishing standardized evaluation frameworks and addressing regulatory requirements will be essential for clinical translation.

## 5. Limitations

Our most critical limitation is using Heart Team decisions as the reference standard without clinical outcome validation. We measured whether the LLM agreed with expert consensus, not whether stratification decisions were clinically correct or resulted in improved patient outcomes. Without prospective follow-up data on mortality, procedural complications, or long-term survival, we cannot determine whether LLM concordance translates to clinical benefit. A system achieving 90% agreement with expert decisions could still be systematically incorrect if those expert decisions themselves were suboptimal. Future validation must assess whether LLM-based stratification predicts actual clinical outcomes in prospective cohorts. Our findings demonstrate the technical feasibility of replicating expert opinion but cannot establish clinical validity or safety for patient care decisions.

GPT-4o’s proprietary training data and closed-source architecture raise fundamental questions about reproducibility. OpenAI does not disclose the specific medical literature, clinical guidelines, or case examples used to train GPT-4o, making it impossible to verify what information the model draws upon. Different research teams using different API versions, regions, or time periods may obtain divergent stratification outcomes for identical patients. Unlike transparent algorithmic approaches, where every calculation step can be audited, the reasoning pathway of commercial LLMs remains opaque, precluding meaningful error analysis when incorrect stratifications occur.

The single-center design limits generalizability to institutions with different decision-making processes, patient populations, and documentation practices. Site-specific adaptation of OCR processing and data extraction prompts would likely be required for reliable function in different healthcare systems. We did not assess inter-rater variability among individual Heart Team members, limiting our ability to determine whether LLM concordance represents alignment with unanimous expert opinion or contested majority view. Our methodology relied on structured clinical vignettes, excluding imaging data rather than raw clinical documents, which may not capture real-world complexity. Potential biases embedded in LLM training data could perpetuate healthcare disparities. Finally, we did not evaluate the real-world impact on clinical team dynamics, workflow efficiency, or patient outcomes, factors that will ultimately determine clinical utility regardless of technical performance metrics.

## 6. Conclusions

LLMs can accurately calculate EuroSCORE II with lower mean absolute error compared to algorithmic approaches, offering a reliable automated method for risk score computation. For operative risk stratification, a commercial LLM achieved 90% concordance with Heart Team decisions using guideline-integrated reasoning. However, the black-box nature of commercial LLMs raises critical reproducibility concerns, and clinical outcome validation is essential before any application can be considered.

## Figures and Tables

**Figure 1 jcm-14-08304-f001:**
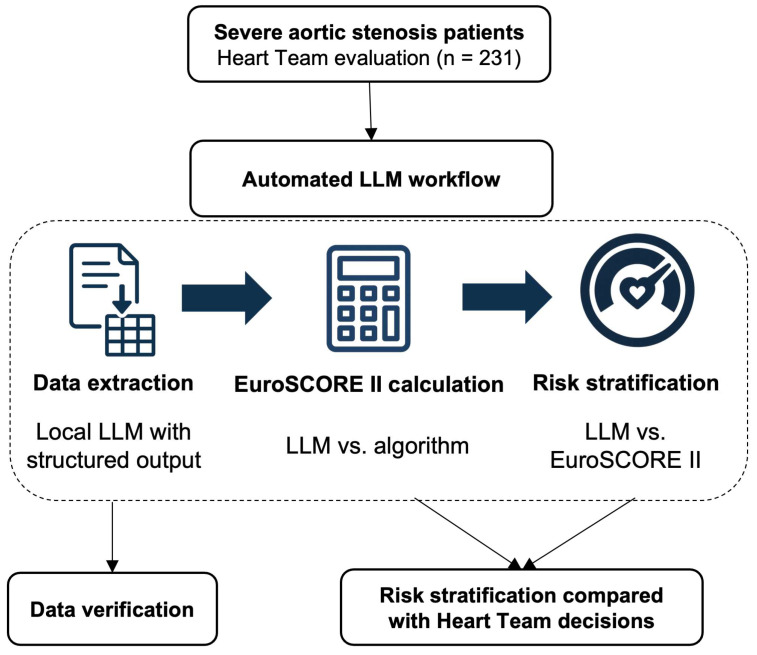
**Study flowchart**. Overview of the automated LLM workflow for severe aortic stenosis risk stratification. The process included data extraction using structured output LLM, EuroSCORE II calculation (LLM vs. algorithmic), and risk stratification (guideline-integrated LLM vs. EuroSCORE-based). All data were manually verified before comparison with Heart Team decisions (n = 231). LLM, large language model.

**Figure 2 jcm-14-08304-f002:**
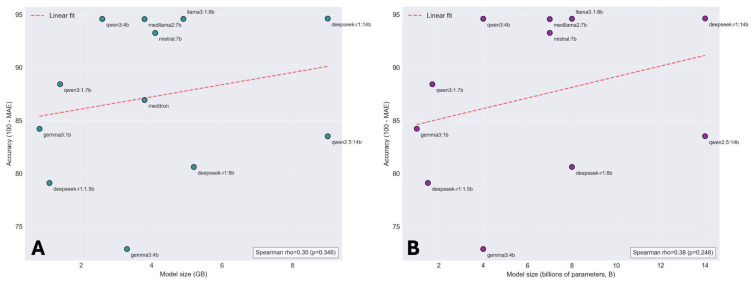
**Local model accuracy for EuroSCORE II calculation according to model size and parameter count**. (**A**) Scatter plot showing the relationship between model size in gigabytes (GB) and accuracy (defined as 100 minus mean absolute error) for EuroSCORE II calculation across 12 open-source language models. The red dashed line indicates linear regression fit. (**B**) Scatter plot showing the relationship between model parameter count (billions of parameters) and accuracy for the same models. The red dashed line indicates linear regression fit. LLM, large language model.

**Figure 3 jcm-14-08304-f003:**
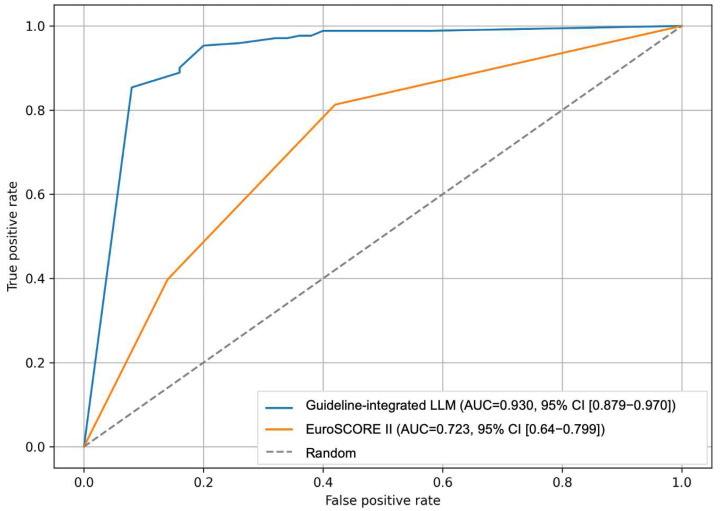
**Accuracy performance of the guideline-integrated LLM compared to the EuroSCORE II method**. Comparison of guideline-integrated LLM (AUC = 0.930, 95% CI [0.879–0.970]) versus EuroSCORE II (AUC = 0.723, 95% CI [0.641–0.799]) for risk stratification. The guideline-integrated LLM showed superior discriminative performance compared to EuroSCORE II-based classification (n = 231). AUC, area under the curve; CI, confidence interval; LLM, large language model.

**Table 1 jcm-14-08304-t001:** Characteristics of the patients at baseline (n = 231).

Characteristic	Value
Demographic	
Female sex—no./total no. (%)	135/231 (58.4)
Age—yr	79.5 ± 7.7
Medical history	
Mean body mass index—kg·m^2^ †	27.4 ± 5.4
Mean EuroSCORE II ‡	4.35 ± 3.15
Mean STS §	2.8 ± 1.62
New York Heart Association class—no. (%)	
I	54 (23.4)
II	105 (45.4)
III	51 (22.5)
IV	19 (8.2)
Syncope linked to aortic stenosis—no. (%)	7 (3.0)
Previous coronary artery disease—no. (%)	144 (62.3)
Previous acute coronary syndrome—no. (%)	23 (9.9)
Previous cardiac surgery—no. (%)	
Coronary artery bypass surgery (CABG)	19 (8.4)
Aortic valve surgery	8 (3.8)
CABG and aortic valve surgery	3 (1.5)
Baseline echocardiogram	
Left ventricular ejection fraction—%	57.8 ± 11.5
Aortic valve surface—cm^2^	0.79 ± 0.24
Aortic valve regurgitation—no. (%)	
Moderate	26 (11.5)
Severe	6 (2.6)
Aortic valve gradient—mmHg	36.9 ± 11.2
Mitral valve regurgitation—no. (%)	
Moderate	33 (14.2)
Severe	3 (1.3)
Tricuspid valve regurgitation—no. (%)	
Moderate	15 (6.4)
Severe	4 (1.7)
Systolic pulmonary artery pressure—mmHg	44.2 ± 14.1
Baseline electrocardiogram—no. (%)	
Atrioventricular block	30 (12.9)
Right bundle branch block	18 (7.7)
Left bundle branch block	28 (12.1)
Atrial fibrillation	56 (24.4)
Cardiovascular disease risk factors—no. (%)	
Diabetes	65 (28.1)
Requiring insulin	17 (7.4)
Hypertension	170 (73.6)
Dyslipidemia	110 (47.6)
Current or previous smoking	39 (16.8)
Non-cardiac previous history	
Chronic obstructive pulmonary disease—no. (%)	18 (7.6)
Moderate kidney disease—no. (%)	69 (29.8)
Severe kidney disease—no. (%)	65 (28.8)
Creatinine—μmol/L	97.7 ± 44.5
Creatinine clearance—mL·min^−1^·1.73·m^−2^	58.1 ± 24.8
Peripheral artery disease—no. (%)	30 (12.9)
Previous vascular surgery—no. (%)	9 (3.8)
Previous stroke or transient ischemic attack—no. (%)	26 (11.4)
Heart Team decision—no. (%)	
Transcatheter aortic valve implantation	168 (72.7)
Surgical aortic valve replacement	50 (21.6)
Medical management	13 (5.6)

Values are mean ± standard deviation or number (%). For continuous variables, the median and interquartile range are presented for non-normally distributed variables. Adapted from Ref. [17]. † The body-mass index is the weight in kilograms divided by the square of the height in meters. ‡ The values on the European System for Cardiac Operative Risk Evaluation II (EuroSCORE II) range from 0 to 100%, with higher scores indicating a greater risk of in-hospital death. § The Society of Thoracic Surgeons Short-term score ranges from 0 to 100%, with higher scores indicating a greater risk of death within 30 days after the procedure.

**Table 2 jcm-14-08304-t002:** Automated workflow processing times.

Workflow Component	Mean Time (Seconds)	Standard Deviation
Data extraction	14.3	3.0
EuroSCORE II calculation	8.1	2.2
Risk stratification	9.4	2.9
**Total workflow**	**32.6**	**6.4**

Values represent mean processing time per patient (n = 231) with standard deviation.

**Table 3 jcm-14-08304-t003:** Receiver operating characteristic analysis.

Approach	AUC	95% CI	Sensitivity	Specificity
Guideline-integrated LLM without EuroSCORE	0.93	0.87–0.97	0.98	0.64
Guideline-integrated LLM with EuroSCORE	0.76	0.69–0.83	0.88	0.64
EuroSCORE-based (non-high risk)	0.72	0.66–0.79	0.98	0.46
EuroSCORE-based (high risk)	0.63	0.57–0.69	0.40	0.86

Values are area under the curve (AUC), 95% confidence interval (CI), sensitivity, and specificity for each risk stratification approach.

## Data Availability

The data presented in this study are available upon request from the corresponding author due to Switzerland’s national law regarding data protection.

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
