# Peer review of "AI-Automated Risk Operative Stratification for Severe Aortic Stenosis: A Proof-of-Concept Study"

_jcm, 2025, doi:10.3390/jcm14238304_

Round 1
Reviewer 1 Report
Comments and Suggestions for Authors
This study presents an innovate proof-of-concept for automating heat team risk assessment in severe aortic stenosis using LLMs. The topic is important in clinical region of utilizing LLMs, it is also original in methodology design. The manuscript is generally well organized and readable.
However, the study has limitation in validation design, potential bias, and incomplete ethical reporting. I would suggest authors consider these issues below before the manuscript acceptance.
- Validation Design. Page 3, Line 107-118: "We evaluated two distinct approaches against heart team decisions as the reference standard..." The validation depends entirely on heart team decisions, which also guide the model's inputs. This introduces circulars reasoning and inflates concordance. Please include or plan an external or outcome-based validation set.
- Potential Bias. Page 8, Lines 277-283:"The counterintuitive finding that excluding EuroSCORE II... suggests anchoring bias". The interpretation is speculative. Please rephrase it.
- Page 10, Lines 333-341. Institutional Review Board Statement is not added.
Author Response
Comment: "Page 3, Line 107-118: 'We evaluated two distinct approaches against heart team decisions as the reference standard...' The validation depends entirely on heart team decisions, which also guide the model's inputs. This introduces circular reasoning and inflates concordance. Please include or plan an external or outcome-based validation set."
Response: We thank the reviewer for raising this important methodological concern. We wish to respectfully clarify that the LLM processes patient clinical data independently without access to the Heart Team's actual decisions. The automated workflow receives only clinical information such as demographics, comorbidities, echocardiographic measurements, and laboratory values along with published guidelines, then generates its own risk stratification. This recommendation is subsequently compared to the Heart Team decision, preventing circular reasoning.
Nevertheless, we fully appreciate the reviewer's underlying concern about validation design limitations. We have made several important revisions to address this issue. The Methods section now includes comprehensive documentation of our Heart Team's composition, meeting structure, and consensus-based decision-making process to provide transparency about how the reference standard was established. The Limitations section explicitly acknowledges that using Heart Team decisions as the reference standard represents a fundamental limitation, as these consensus decisions lack long-term outcome validation and do not represent objective truth. We clarify that concordance with Heart Team decisions does not guarantee accuracy in predicting actual patient outcomes.
The Discussion section now includes detailed proposals for future prospective validation studies that assess whether LLM-based stratification predicts actual procedural mortality, morbidity, and long-term survival. Such outcome-based validation will be essential to determine clinical utility beyond replication of expert consensus. We have noted in the manuscript that we are currently conducting an external validation study of our pipeline in clinical practice.
Comment: "Page 8, Lines 277-283: 'The counterintuitive finding that excluding EuroSCORE II... suggests anchoring bias'. The interpretation is speculative. Please rephrase it."Response: We have revised the Discussion to present multiple possible explanations for this counterintuitive finding rather than presenting anchoring bias as the definitive mechanism. The revised text now acknowledges that the improved performance when excluding EuroSCORE II may reflect several possibilities: numerical risk scores creating anchoring bias that constrains holistic assessment, LLM limitations in processing mixed numerical and textual inputs, or guideline-based reasoning better capturing the multifactorial nature of clinical decision-making. We explicitly acknowledge the speculative nature of these interpretations.
Reviewer 2 Report
Comments and Suggestions for Authors
Overall Impression
This manuscript presents an ambitious, timely proof-of-concept: automating the Heart Team’s operative risk-stratification workflow for severe aortic stenosis using LLMs. The idea is revolutionary in spirit… it aims not only to speed tasks, but to reframe multidisciplinary decision support as a reproducible, auditable pipeline. That said, the paper reads simultaneously like a polished technical demonstration and an early prototype: it convincingly shows feasibility (speed, concordance metrics) but needs stronger systematic framing (limitations, reproducibility steps, and clearer boundary conditions) before it can be read as a field-changing, generalizable solution.
Novelty: similar studies and preprints exist (LLM applications to Heart Team decisions, AI risk stratification for TAVR), but your work adds a concrete integrated pipeline (OCR → structured extraction → LLM score computation → guideline-integrated LLM stratification) with a reasonable sample (n=231) and a direct comparison of “LLM w/ and w/o EuroSCORE” which is a useful experimental contrast. However, several recent systematic/review/empirical papers (2023–2025) address overlapping questions and at least one recent systematic review on AI in TAVR risk prediction (Shojaei et al. 2025 https://doi.org/10.3390/jpm15070302 ) and multiple cohort studies of LLMs assisting Heart Teams should be acknowledged and explicitly contrasted with your claims of “first” or “novelty.”
Recommendation: Tone the “first ever” implication down slightly in the Introduction/Discussion; instead emphasize “comprehensive integration of steps and choice-experiment of including vs excluding numeric anchoring (EuroSCORE)” as your unique contribution.
- The manuscript’s present phrasing is defensible if the authors mean “no prior study has attempted the entire automated pipeline starting from raw clinical files (OCR) through to guideline-integrated Heart Team stratification.” If that is the intended, precise scope, keep the claim but make it explicit and cite related partial-scope studies so readers see the difference.
- If the authors intend a broader claim (for example, “no study has used LLMs for Heart Team decisions at all”), that would be incorrect — multiple small/curated studies exist and should be acknowledged.
Suggested changes: “While recent studies have explored the use of LLMs for Heart Team decision support or ____, no prior study — to our knowledge — has automated the entire Heart Team evaluation pipeline starting from ___. This study therefore___”
STRENGTH
This is a highly prepared manuscript with:
- The methods are described stepwise (Step 1–4) and authors provided code/supplement link — excellent for reproducibility. (lines 80–91).
- Statistical methods are appropriate (Bland–Altman, ICC, ROC/AUC, bootstrap) and reported clearly (lines 127–141).
WEAKNESS/REQUIRED CLARIFICATIONS
- Data provenance & selection bias: the Heart Team culture and single-center nature are acknowledged (lines 300–305), but authors should be explicit about the criteria used in Heart Team decision (weighting between operative risk, frailty, imaging cues, patient preference), and whether Heart Team decisions were unanimous or by majority ( this matters because your primary endpoint is concordance with that decision ). Please report inter-rater variability among Heart Team members or whether a ground-truth consensus was used (not simply “the Heart Team decision”). (Lines 66–74 and 300–319).
- Ground truth limitations: reiterate that Heart Team decision ≠ patient outcome and avoid language implying equivalence (several sentences in Abstract/Discussion risk this; see Abstract lines 22–33 and Discussion lines 224–241). Replace any phrasing that equates “concordance with Heart Team” with “agreement with Heart Team at decision time.”
- OCR / data missingness handling: the hallucination issue (lines 258–271 and 156–161) is a major practical limitation; a fuller description of how missing/ambiguous fields were flagged, whether the LLM had an explicit “I don’t know / missing data” response option, or if an uncertainty score was returned should be added. Present a short protocol/flowchart for safe-failure modes (e.g., when OCR confidence < X, abort and flag to human). This is critical for clinical translation.
- External validation and prospective plan: the Discussion (lines 290–298) mentions prospective validation. Strengthen by proposing concrete next steps (multi-center validation, inclusion of imaging DICOM data, clinician acceptability studies, and outcomes linkage). State a minimum dataset that must be present for the pipeline to be reliably run (e.g., images + echo values + key labs).
GRAMMAR & ORDER
- Results — numeric presentation & tense:
Quote (lines 150–156): “All 231 patient dossiers were successfully processed without technical failures. The complete automated workflow achieved mean processing time of 32.6±6.4 seconds per patient. Data extraction required 14.3±3.0 seconds, EuroSCORE calculation took 8.1±2.2 seconds, and risk stratification required 9.4±2.9 seconds.” Suggestion: consider presenting the times in a small table for clarity; use consistent tense and units. - Repetitive rhetorical phrases:
Several paragraphs restate the same limitation (hallucinations/OCR) multiple times (e.g., lines 158–161, 258–271, 319–323). Consolidate into one dedicated “Safety & hallucinations” subsection that lays out evidence, frequency, examples, and mitigation strategies. - Wordiness/metaphore: Phrases like “unprecedented opportunities,” “fast processing time represents a substantial reduction,” “experimental and requires extensive validation before any thought of clinical deployment” appear across Introduction/Discussion (e.g., lines 53–63, 231–241). Make them crisper to improve scientific tone.
- Repeated ideas / structural moves (what to remove or relocate) -Move general AI background (lines 53–66) to a short paragraph; the introduction is currently slightly long and repeats global AI promises later in Discussion. Keep intro focused: clinical problem → gap → study aim.
- Consolidate all hallucination/OCR text into Methods/Results/Limitations rather than repeating examples in Results and then again in Discussion.
-The sentences on “model size not correlating with accuracy” (lines 283–289) are important; move one clear sentence of interpretation into the Discussion’s core findings and remove redundant exposition.
Novelty Gap:
- Generazibility beyond one single institution: The study uses data from one institutional Heart Team. Could the authors elaborate on how institutional differences in Heart Team composition, local workflow, or documentation style might affect the model’s reproducibility? For instance, would the same OCR–LLM pipeline function reliably in another hospital using different EHR templates or languages, or would it require site-specific re-prompting or re-training?
- How do the authors envision the integration of this automated system into real clinical decision pathways? Specifically, who would bear clinical responsibility if the automated EuroSCORE or guideline-based recommendation diverged from a physician’s judgement ; and what human-in-the-loop safeguards are proposed before deployment in practice?
- Given that surgical risk models and guideline criteria evolve rapidly (e.g., ESC/EACTS updates every few years), how will the pipeline adapt to these changes? Are there plans for a “living” or self-updating version of the workflow, or would the authors expect periodic manual retraining and prompt revision to prevent model drift and outdated recommendations?
Verdict / recommendation
This is an important, timely, and well-executed proof-of-concept, revisions focused on (1) clearer framing of novelty against recent LLM/Heart-Team literature, (2) consolidating and precisely documenting hallucination/OCR failure modes with mitigation strategies, (3) adding the exact extraction variable list and Heart Team decision process details, and (4) a careful editorial pass for concise scientific English.
Author Response
Comment: "Novelty: similar studies and preprints exist (LLM applications to Heart Team decisions, AI risk stratification for TAVR), but your work adds a concrete integrated pipeline (OCR → structured extraction → LLM score computation → guideline-integrated LLM stratification) with a reasonable sample (n=231) and a direct comparison of 'LLM w/ and w/o EuroSCORE' which is a useful experimental contrast. However, several recent systematic/review/empirical papers (2023–2025) address overlapping questions and at least one recent systematic review on AI in TAVR risk prediction (Shojaei et al. 2025 https://doi.org/10.3390/jpm15070302) and multiple cohort studies of LLMs assisting Heart Teams should be acknowledged and explicitly contrasted with your claims of 'first' or 'novelty.'"Response: We have substantially restructured and condensed the Introduction to provide more appropriate context while clearly defining our specific contribution. The revised Introduction now explicitly acknowledges the Shojaei et al. 2025 systematic review and the broader literature on LLM applications for cardiovascular risk assessment and Heart Team decision support. We have removed absolute claims such as "no study has yet attempted" and replaced them with measured language that acknowledges existing work while precisely defining our contribution as the automation of the complete integrated pipeline from unstructured clinical documents through optical character recognition, structured data extraction, risk score calculation, and guideline-integrated stratification. The Introduction is now more focused and concise, following the recommended structure of clinical problem, current limitations, AI opportunity, gap in literature, and study aims, without overstating novelty claims.
Response: We have substantially expanded the Methods section to provide comprehensive detail about our Heart Team's decision-making process. The revised section now specifies the team's composition including interventional cardiologists, cardiac surgeons, cardiac anaesthesiologists, and imaging specialists, describes the weekly meeting structure using standardized protocols, and details the consensus-based decision-making process that considers operative risk scores, frailty assessment, echocardiographic findings, coronary anatomy, patient preferences, and procedural feasibility. We explicitly state that in cases where initial opinions diverged, discussion continued until consensus was reached, and the final consensus decision was recorded as the reference standard for this study.
In the Limitations section, we have added acknowledgment that we did not systematically assess inter-rater variability among individual Heart Team members before consensus was reached, and therefore cannot quantify the degree of initial disagreement or the factors that influenced final consensus decisions. We note this limits our ability to determine whether the LLM's concordance represents alignment with unanimous expert opinion or with a potentially contested majority view.
Response: We have conducted a comprehensive review of language throughout the manuscript, particularly in the Abstract, Discussion, and Conclusions sections. We have systematically replaced terms that could imply objective accuracy such as "accuracy" and "correctly classified" with more appropriate terminology including "concordance," "agreement," and "alignment with Heart Team decisions." The Limitations section now includes explicit statement that Heart Team decisions represent expert consensus at the time of evaluation rather than objective truth or patient outcomes, and that concordance with these decisions does not guarantee accuracy in predicting actual patient outcomes or appropriateness of intervention choice.
Comment: "OCR / data missingness handling: the hallucination issue (lines 258–271 and 156–161) is a major practical limitation; a fuller description of how missing/ambiguous fields were flagged, whether the LLM had an explicit 'I don't know / missing data' response option, or if an uncertainty score was returned should be added. Present a short protocol/flowchart for safe-failure modes (e.g., when OCR confidence < X, abort and flag to human). This is critical for clinical translation."
Response: We have conducted additional statistical analyses to rigorously quantify the hallucination problem and substantially expanded the relevant sections throughout the manuscript. In the Results section, we have added formal statistical testing using McNemar's test demonstrating overall extraction accuracy of 94.7% across 38 variables, but revealing significant systematic overestimation for hypertension (38 cases representing 16.4% overestimation with 95% CI 10.1-22.7, p less than 0.001) and dyslipidemia (75 cases representing 32.5% overestimation with 95% CI 25.3-39.7, p less than 0.001). We report that accuracy improves to 98.9% for the remaining 36 variables when these hallucinated variables are excluded.
In the Discussion section, we have proposed a detailed safe-failure protocol comprising four specific components: requiring OCR confidence scores above 0.90 for processing to proceed, implementing automatic flagging of contradictory or ambiguous values as missing data, utilizing structured output schemas that force selection between defined categories including uncertain options, and incorporating calibrated confidence scores for all extracted variables. We explicitly acknowledge that our current proof-of-concept implementation lacks automated uncertainty detection capable of identifying ambiguous OCR output or distinguishing hallucination from genuine extraction, representing a critical limitation that must be addressed before any clinical application. We have also acknowledged that the hallucinations occurred despite explicit prompting strategies designed to prevent such behavior, noting that all prompts specifically instructed models to return missing or uncertain values when data was absent or ambiguous.
The discussion of hallucinations has been consolidated to eliminate redundancy across sections, with each mention now serving a distinct purpose: quantitative statistical evidence in Results, mechanistic interpretation and proposed mitigation strategies in Discussion, and broader implications in Limitations.
Comment: "External validation and prospective plan: the Discussion (lines 290–298) mentions prospective validation. Strengthen by proposing concrete next steps (multi-center validation, inclusion of imaging DICOM data, clinician acceptability studies, and outcomes linkage). State a minimum dataset that must be present for the pipeline to be reliably run (e.g., images + echo values + key labs)."
Response: We have substantially restructured and expanded the Discussion section to provide comprehensive and concrete proposals for future validation. The revised section now specifies that prospective studies should assess whether LLM-based stratification predicts actual procedural mortality, morbidity, and long-term survival rather than merely concordance with expert opinion. We detail the need for multi-center external validation across different Heart Team cultures and documentation styles to establish generalizability, particularly emphasizing that institutional variations in electronic health record templates, language, and clinical workflows may require site-specific adaptation of OCR processing and data extraction prompts.
The revised Discussion now explicitly proposes integration of multimodal data including DICOM imaging and direct echocardiographic parameter extraction to enhance system comprehensiveness and address current limitations from relying solely on structured clinical vignettes. We have specified minimum dataset requirements that should include patient demographics, complete echocardiographic measurements, laboratory values, documented comorbidities, and prior cardiac interventions to ensure reliable pipeline operation.
We have also added proposals for clinician acceptability studies evaluating trust, workflow integration, and human-AI collaboration dynamics to understand real-world adoption barriers, as well as health economic evaluation comparing costs and outcomes of AI-assisted versus traditional workflows. This section now acknowledges evidence that high accuracy does not guarantee cost-effectiveness, with improving AI sensitivity potentially increasing medical costs while increasing specificity reduces unnecessary referrals but may weaken detection capability.
Response: We have created a new Table 2 titled "Automated workflow processing times" that presents all timing data in a clear, organized format. The table displays individual component times for data extraction, EuroSCORE II calculation, and risk stratification, along with the total workflow time. All values are presented as mean with standard deviation in consistent units of seconds. The Results text has been revised to concisely reference this table rather than listing all values in prose format, improving both clarity and readability.
Response: We have comprehensively restructured the discussion of hallucinations throughout the manuscript to eliminate all redundancy while ensuring complete coverage of this critical safety concern. Each section now addresses hallucinations with a distinct purpose and without repetition. The Results section presents rigorous quantitative statistical evidence using McNemar's test with specific frequencies and confidence intervals. The Discussion section provides mechanistic interpretation of why hallucinations occur, contextualizes the problem within the broader medical AI literature, and proposes detailed mitigation strategies through the four-point safe-failure protocol. The Limitations section addresses the broader implications for real-world implementation. This reorganization ensures that the hallucination issue receives appropriate emphasis as a fundamental safety concern while maintaining scientific rigor and avoiding repetitive presentation.
Response: We have systematically revised verbose and metaphorical language throughout the Introduction and Discussion to achieve greater scientific precision and conciseness. We have eliminated imprecise expressions and replaced them with direct, quantitative descriptions of findings and their implications. The revised text maintains appropriate scientific tone throughout while improving clarity and readability.
Response: We have implemented comprehensive structural revisions to eliminate redundancy and improve logical flow throughout the manuscript. The Introduction has been condensed and restructured to follow the recommended progression of clinical problem, limitations of current approaches, AI opportunity, gap in literature, and study aims. General AI background has been compressed into a focused paragraph that directly connects to study objectives, removing repetitive discussion of AI promises that previously appeared in later sections.
The discussion of hallucinations has been completely restructured as detailed in our response to Comment 2.7, with each section serving a distinct non-overlapping purpose. The discussion of model size and accuracy has been streamlined into a concise paragraph that presents the finding and its implications without redundant exposition, noting that smaller models performed comparably to larger ones and suggesting that domain-specific optimization and structured clinical data may matter more than raw scale for this type of medical task.
Response: We have added comprehensive discussion of generalizability concerns to the Limitations section. The revised text explicitly acknowledges that the single-center design limits generalizability to institutions with different decision-making processes, patient populations, and documentation practices. We state directly that site-specific adaptation of OCR processing and data extraction prompts would likely be required for reliable function in different healthcare systems, given institutional differences in documentation style, electronic health record templates, language, and clinical workflows. We note that the degree of adaptation required and the feasibility of creating a truly institution-agnostic system remain open questions requiring multi-center validation.
Response: We have added detailed discussion of clinical integration, responsibility, and ethical frameworks to the Discussion section. The revised text emphasizes that any deployment must function exclusively as decision support rather than autonomous decision-maker, with final responsibility remaining with treating physicians and the Heart Team. We propose specific human-in-the-loop safeguards including mandatory physician review of all LLM-extracted data, risk calculations, and stratification recommendations before clinical decisions, with clear documentation of physician agreement or disagreement with automated suggestions.
We acknowledge that the legal and ethical framework for clinical responsibility when using AI-assisted tools remains an evolving area requiring regulatory guidance, institutional policy development, and clear communication with patients about the role of AI in their care. The Limitations section now includes acknowledgment that we did not evaluate real-world impact on clinical team dynamics, workflow efficiency, decision-making responsibility, or patient outcomes, factors that will ultimately determine clinical utility regardless of technical performance metrics.
Response: We have integrated discussion of guideline adaptation into the future validation section of the Discussion. The revised text identifies exploration of systems that automatically update their decision logic when ESC/EACTS guidelines are revised or new evidence emerges as an important research direction. We emphasize that such adaptive systems would require regular human oversight and revalidation to ensure safe incorporation of updated criteria without introducing errors from outdated or misinterpreted recommendations. This framing positions guideline adaptation as a critical area for future research while acknowledging the safety considerations inherent in automated updating systems.